# THE PRINCIPLE OF MAXIMUM ENTROPY AS A TOOL FOR UNSTRUCTURED DATA PROCESSING

## ABSTRACT

The maximum entropy principle (MaxEnt) offers several advantages that make it suitable for use in unstructured data processing. However, finding the MaxEnt distribution requires solving an optimization problem using Lagrange multipliers with minimal prior data. While most studies rely on the well-known moment problem, we examine only first- and second-order moments. From the perspective of the calculus of variations, it has been shown that the maximum entropy distribution for a known first-order moment (the mathematical expectation) is a Gibbs distribution with an associated exponential function. For the second-order moment (the variance), MaxEnt is a convex function whose extremum region achieves the greatest information "saturation". This study demonstrates the effectiveness of this approach in digital image processing for identifying color contrast zones that most accurately capture silhouettes and object data. Knowledge of central moments provides additional information about texture, reproduced objects, and image elements. The effectiveness of the maximum entropy principle is demonstrated in applications to supervised learning. Further research focuses on generalizing the principle to f-entropy (specifically, with respect to moment problems for Rényi and Tsallis entropy), and on applied evaluation of the principle's effectiveness on time series in comparison with recurrent artificial neural network technologies.

## 1 INTRODUCE

According to Janes, the maximum entropy principle states that if nothing is known about a distribution, the distribution with the highest entropy should be chosen as the most preferable by default Jaynes (1968). Given Laplace's principle of indifference, such a distribution would be a uniform law $\mathcal{H}(X) : \left\{ unif\left(\bar{x}, \sigma_x^2\right), \forall p\left(\bar{x}, \sigma_x^2\right) \in \mathbb{R}\right\}$, where $unif\left(\bar{x}, \sigma_x^2\right)$. is uniform distribution with central moments of first and second order, respectively. Thus, if none of the possible a priori solutions cannot be called more probable, then the entropy of this distribution will be maximum Niven & Andresen (2009).

The essence of MaxEnt in unstructured data processing is quite twofold. The one hand, the MaxEnt distribution minimizes the Hessian matrix $p^*(X) \to \mathcal{H}(X)$ on the manifold $\mathbb{R}^{m \times n}$. This is a useful property of image or signal processing, since the PDF projection allows you to specify the original denoised data by filtering it out Baggenstoss (2015). On the other hand, this leads to an increase in the loss function $L : \left\{x_1, \ldots, x_n \mid \mathcal{X}\right\}$ on the manifold $\mathbb{R}^{m \times n}$. Hence, the accuracy of processing decreases. Nevertheless, the MaxEnt of the distribution in the presence of a number of restrictions is an important tool for understanding the properties of unstructured data processing (Fig. 1).

Maximized entropy is quite often used when information about the "behavior" of the medium is known. Initially, the application of this principle was found in thermodynamic physics. However, after the works of T. Janes, its applications in applied information theory began to be actively developed Jaynes (1968); E. T. Jaynes (1957). The study of the MaxEnt principle is often associated with the Bayesian statistical approach Kim et al. (2018); Caticha (2021); Grünwald & Dawid (2004). There is a group of approaches where MaxEnt is investigated by statistical tools for image classification and prediction based on learning data Mazuelas et al. (2022); Qiu et al. (2017); Yin et al. (2018). At the same time, a number of papers Hong & Schonfeld (2008); Nunez & Llacer (1989) aim on the

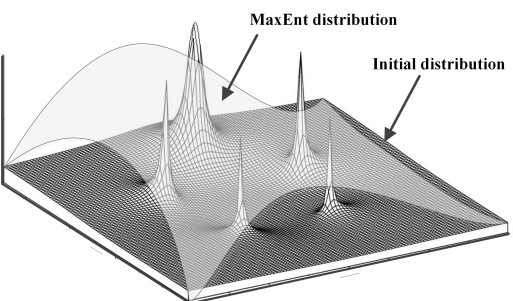

Figure 1: MaxEnt distribution for separated multimode distribution with known central moments

processing conditions of learning data reconstruction images. It is known MaxEnt principle can be used in processing of radar information and thermal vision Kvasnov (2023b;a; 2025).

Among the variety of works there is no detailed analysis of the MaxEnt distribution on finite statistical moments. We believe that the applied nature of MaxEnt should take into account these factors, in particular, the metric for some arbitrary mean. The purpose of our study is to try to estimate how a priori distribution parameters affect the properties of the MaxEnt distribution in unstructured data processing.

## 2 ENTROPY AS A MEASURE OF DYNAMIC ESTIMATION OF THE DISTRIBUTION

The concept of entropy is denoted as a measure of diversity. For the case of the MaxEnt principle, it is necessary to find some "optimal" distribution satisfying the given a priori central tendency data $\Upsilon_{\mathcal{P}} \in \mathcal{L}^{\mathcal{P}}$. Generally, for a discrete random variable distributed on $X \subseteq \mathbb{R}^n$ $(n < \infty)$ on the condition $X = \{x\}_{i=1}^n : p(x) > 0$, the information entropy can be defined through Rényi or Tsallis formalism Ghosh & Basu (2021). The Rényi entropy can be written as

$$H_\alpha(X) \stackrel{def}{=} \frac{1}{1-\alpha} \log \sum_{i=1}^n p_i^\alpha,\tag{1}$$

where $\alpha \geq 0$ is the entropy coefficient. In fact, we investigate the behavior of the MaxEnt distribution as a function of the Hölder moving mean. In this case, we obtain a family of distributions with characteristic properties. To solve this problem, we use the well-known approach based on Lagrange multipliers Popkov (2021). We have already proven the theorem Kvasnov et al. (2023).

**Theorem 2.1.** *If $p^*(X \mid T(x))$ is a distribution density with sufficient statistic $T(x_1, \ldots, x_n)$ and a known unique mean (Hölder mean) $\mathcal{M}_\ell(X)$ on the area $\ell \in [2, +\infty)$, then the upper and lower bounds of the distribution density of the maximized entropy $\mathcal{H}\{p^*(X) \mid \mathcal{M}\}$ $\forall X \in \mathbb{R}$ are:*

$$\begin{cases} \sup\{\mathcal{H}(X \mid \mathcal{M}_2)\} := G_\xi(X) \\ \inf\{\mathcal{H}(X \mid \mathcal{M}_{+\infty})\} := unif[p^*(X)] \end{cases}\tag{2}$$

*where $G_\xi(X)$ is the Gibbs distribution with the parameter $\xi \in (0, 1)$.*

The main task is to evaluate the conclusions of Theorem 1 with respect to known distribution laws and original data. The MaxEnt distribution was checked for different values of the Hölder metric $\mathcal{M} : \{-\infty < \ell < \infty\}$. The sample size contained 100 observations for the area $\mathbb{R}^2$. Since numerical analysis at $\ell \to \pm\infty$ requires serious computational resources, we limited ourselves to $\sup(\ell) = 20$ and $\inf(\ell) = -20$. Below are the joint scatter plots of the a priori distribution density $p^*(X)$ and density of MaxEnt distribution $\mathcal{H}(\cdot \mid \mathcal{M}_\ell)$ in two-dimensional space $\mathcal{L}^{\mathcal{P} \in [2, \infty)}$ (Fig. 2).

Next, let's consider the case of a central tendency in space $\mathcal{L}^{\mathcal{P} \in [0, 1]}$, where it is considered on a non-unique mean M. The calculation conditions were similar to those for the series of calculations in fig. 2. The results are shown in Fig. 3.

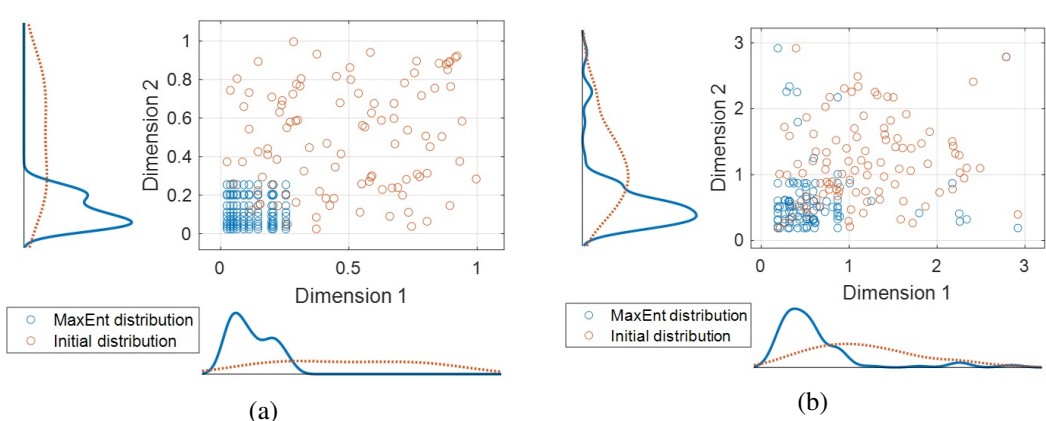

Figure 2: Uniform distribution density Unif $[0,1]$ (on the left) and Rayleigh distribution density Rayl $[0, \infty)$ (on the right) and their corresponding MaxEnt distributions $\mathcal{H}(.)$ for a unique mean $\mathcal{M}_{\ell=0}$ (geometric mean) on manifold $\mathbb{R}^2$ (100 discrete samples)

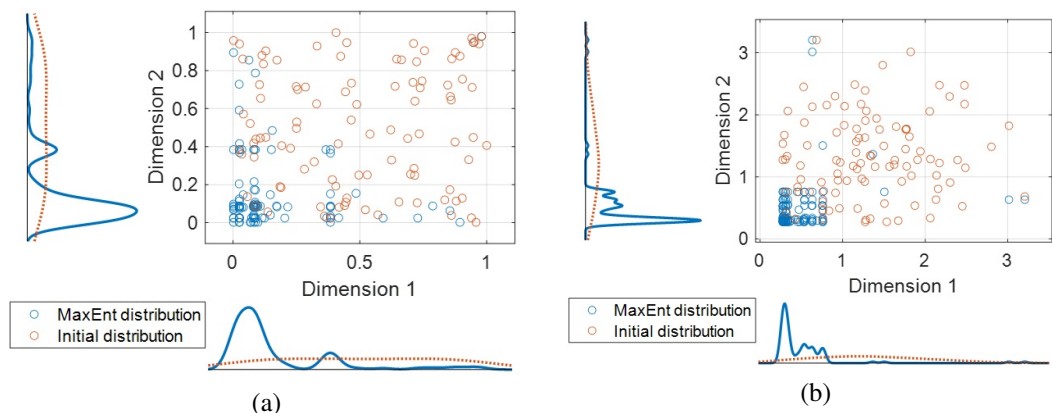

Figure 3: Uniform distribution density Unif $[0,1]$ (on the left) and Rayleigh distribution density Rayl $[0, \infty)$ (on the right) and their corresponding MaxEnt distributions $\mathcal{H}(.)$ for a non-unique mean (median) on manifold $\mathbb{R}^2$ (100 discrete samples)

The MaxEnt distribution density with a known median has a more isochronous character. We explain this by the Van Zwet condition, if $F(med(x_n) - x_i) + F(med(x_n) + x_i) \geq 1 \quad \forall x_i \in X, F(.)$ is the distribution function for the known sample X. On the other hand, a sufficient sample $T(X)$ leads to a localization of the variance $T(x) : \{\mathfrak{D}(x) \to 0, \ mod(x_n) \leq med(x_n)\}$. It is assumed that this behavior of the MaxEnt distribution is more relevant for supervised learning problems Mazuelas et al. (2022).

## 3 MaxEnt Distribution Applications for image processing

Now let's consider the application of Theorem 1 in image processing applications. The unimodal distribution has the property $\mathfrak{D}_{UM}(X) \to \min$, i.e. the set of generalized averages $\mathcal{M} : \{0 \leq \mathcal{P} < \infty\}$ have the minimal variance. In addition, the unimodal distribution satisfies the Van Zwet condition, which restricts the bounds of the MaxEnt distribution. Thus, a priori knowledge of the central tendency reveals bounds of the unequal informativity. The MaxEnt distribution of the grayscale image will be Gibbs distribution with the slope of the function corresponding to the known mean $\mathcal{L}^{\mathcal{P}} : \{Mode\ \mathcal{P} = 0; Median\ \mathcal{P} = 1; Mean\ \mathcal{P} = 2; Midrange\ \mathcal{P} = \infty\}$. The obtained analysis results are shown below (Fig. 4).

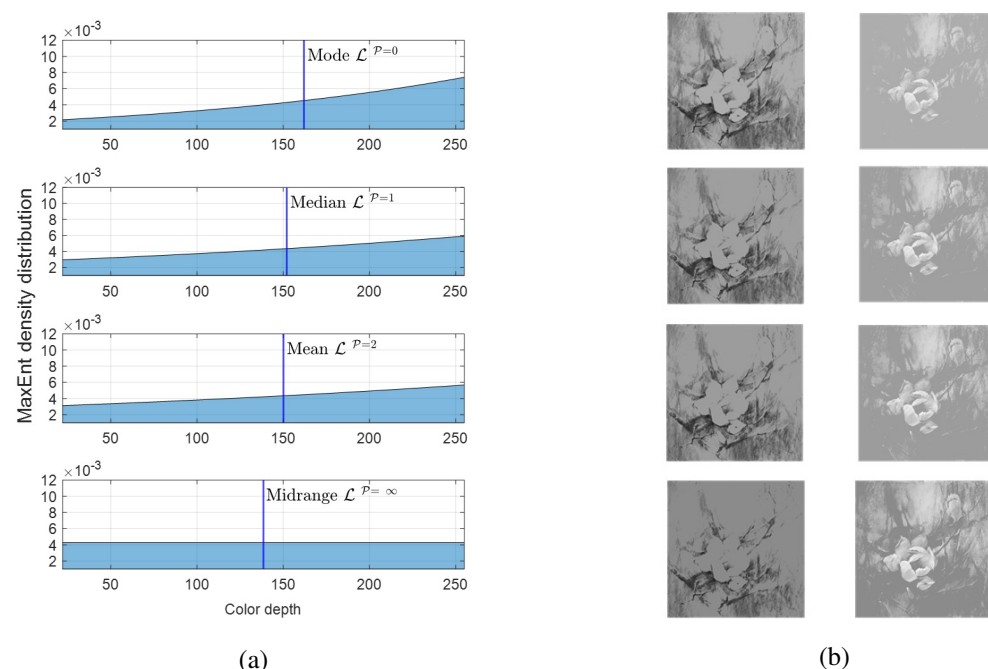

(a)                 (b)

Figure 4: (a) MaxEnt density distribution as a function of color depth for different values of the central tendency $\Upsilon_{\mathcal{P}} : \{0 \leq \mathcal{P} < \infty\}$ at the unimodal distribution; (b) Redistribution of image intensity with respect to 0.5 quantile of MaxEnt distribution for the unimodal distribution with a known CT: (1st row) Mode - $\mathcal{L}^{\mathcal{P}=0}$, (2nd row) - $\mathcal{L}^{\mathcal{P}=1}$, (3rd row) Mean - $\mathcal{L}^{\mathcal{P}=2}$, (4th row) midrange - $\mathcal{L}^{\mathcal{P}=+\infty}$

The shift of the image contrast (Fig. 4b) to the zone of light tones leads to an increase in the density of the MaxEnt distribution, i.e., a loss of the image informativity. This can be clearly seen in Fig. (4a), where the contrast disequilibrium is for a known mode $\mathcal{H}(\bullet | x > \mathcal{M}_0) \gg \mathcal{H}(\bullet | x < \mathcal{M}_0)$, therefore the left image in Fig. 4(b) is more informative than right image in Fig. 4(b). However, there is an "information equilibrium" with respect to the midrange $\mathcal{H}(\bullet | x > \mathcal{M}_\infty) \approx \mathcal{H}(\bullet | x < \mathcal{M}_\infty)$ (Fig. 4a). Thus, the average of mode is more informative on the unimodal distribution

## 4   MAXENT RESULTS ON SUPERVISED LEARNING

Unstructured data can be evaluated by the MaxEnt principle in machine learning tasks. The MaxEnt principle implies smoothing of any distribution. However, even under a unique mean, the principle can perform a partitioning of a set into classes []. The question of finding the best MaxEnt distribution for supervised learning is twofold. On the one hand, it is necessary to choose such a mean (including Hölder mean, weighted mean, etc.), which most accurately describes the convergence of distributions under the conditions of the statistical significance criterion $p^* \{x_1, \ldots, x_n\} \xrightarrow{Test\ statistic} \mathcal{H}\{x_1, \ldots, x_n\}$. On the other hand, the MaxEnt distribution should satisfy the conditions of continuity and smoothness, that is, the Hessian condition $\det\left[\partial^2/\partial x_i \partial x_j \left(\mathcal{H}(x_1, \ldots, x_n)\right)\right] \neq 0 \quad \forall i, j \in \{x_1, \ldots, x_n\}$, so that the parametrization of the distribution is achieved. We conducted a supervised learning study for single-mode distributions on central tendency. As a learning sample, we used data on planes from the reference book [].

The MaxEnt distribution increases the classification accuracy for a limited number of supervised learning methods. The upper bound of the accuracy corresponds to the original probability distribution, except for the Trees model. We cannot strictly explain the class of problems where it is appropriate to use the MaxEnt principle. In this sense, the approach proposed by Mazuelas [9], who proposes to use minimax risk classifiers (MRCs) to estimate the extreme distribution by means of

Table 1: Results of supervised learning for different methods(in percentage)

| Parameter | Model 1: Trees | Linear discriminant | Gaussian naive Bayes | Linear SVM | KNN |
|---|---|---|---|---|---|
| Initial density | 80.3 | 74.9 | 68.0 | 79.0 | 69.4 |
| MaxEnt (Hölder mean $\mathcal{M}_{-\infty}$) | 82.0 | 68.9 | 61.5 | 74.6 | 55.5 |
| MaxEnt (Hölder mean $\mathcal{M}_0$) | 81.1 | 71.3 | 66.1 | 76.6 | 66.9 |
| MaxEnt (Hölder mean $\mathcal{M}_{+\infty}$) | - | - | - | - | - |
| MaxEnt Median | 79.2 | 68.6 | - | 76.2 | 45.9 |
| MaxEnt Mode | 80.1 | 63.7 | 61.5 | 75.7 | 56.6 |

convex optimization appears to be more effective. It seems that MaxEnt on central tendency has no advantages for using in applied tasks, in particular, in learning by precedents.

## 5 CONCLUSION

The paper considers the behavior of the maximized entropy under the condition of a known central tendency of the a priori distribution. In contrast to known works, where MaxEnt is studied with respect to the moment problem, we investigate a different practical approach for application to unstructured data processing, in particular, image processing and supervised learning problems. In this article, we modeled and estimated the maximized entropy for various forms and types of distributions. In particular, we considered the behavior of MaxEnt for uniform and Rayleigh distributions and Gaussian mixture distributions. In applied problems, the MaxEnt principle can be used for image processing and supervised learning. Image processing has shown the efficiency of filtering using MaxEnt on the generalized Hölder mean. Supervised learning on MaxEnt distributions with different a priori means showed no advantage over the original distributions. Further research in the field of applied MaxEnt principle problems is related to the use of different kinds of entropy. In particular, Rényi or Tsallis entropy can advance the solution of the moment problem. In applied problems, deeper attention should be paid to the Gaussian mixture of distributions under supervised learning conditions.

ACKNOWLEDGMENTS

This study was supported by the Ministry of Science and Higher Education of the Russian Federation, state task FFZF-2025-0003

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
