# OpenReview forum: "The principle of maximum entropy as a tool for unstructured data processing"
_mathai.club/MathAI/2026/Conference — 2026 Oral_

### Official Review · Reviewer_V4tZ · 2026-03-11
**Article contains an interesting setting, but in its current form, it is more of an applied/exploratory type than a complete study.**

**Rating:** 4
**Confidence:** 4

**Review:**

The article considers the application of the maximum entropy principle to the processing of unstructured data with known characteristics of the central tendency of the prior distribution. The authors discuss MaxEnt in relation to generalized Holder means, formulate a theoretical statement about the boundaries of the maximum entropy distribution, and then consider applications to image processing and supervised learning tasks.

The strong point of the article is the attempt to consider the behavior of MaxEnt not only in the classical formulation of the moment problem, but also through the constraints associated with central tendency.
It is also positive that the authors do not hide negative or ambiguous results in the applied part, in particular, the lack of a sustainable advantage in supervised learning.

Taking it to the cons, the mathematical part of the article appears to be limited in depth. Although, there is a lack of new statements with proofs, analysis of the conditions of applicability and etc. Also there are empty bibliographic references in the supervised learning section.

Experiments on supervised learning show that the MaxEnt transform in most cases does not improve performance compared to the original distribution; the authors themselves acknowledge the lack of advantage over the original distributions. This makes the paper's overall motivation even less compelling.

---

### Official Review · Reviewer_xo2u · 2026-03-11
**Interesting application idea, but the paper is not rigorous or convincing enough**

**Rating:** 4
**Confidence:** 2

**Review:**

The paper explores the use of the maximum entropy principle for unstructured data processing, with applications to image processing and supervised learning. The topic is potentially interesting, and it is a positive aspect that the paper reports mostly negative results for the supervised learning part instead of overstating its success. However, in its current form the work is not sufficiently rigorous: the main theoretical claims are not clearly justified, several statements are imprecise, and the presentation contains noticeable notational and typographical problems. The empirical validation is also weak, since the image-processing part lacks strong baselines and objective metrics, while the supervised learning results mostly do not show an advantage over the original data representation. Overall, the paper appears exploratory rather than complete and is below the acceptance threshold in its current form.

---

### Decision · Program_Chairs · 2026-03-14

**Decision:**

Accept (Oral)

**Comment:**

Dear Author(s),

On behalf of the Program Committee of the International Conference on Mathematics of Artificial Intelligence (MathAI 2026), we are pleased to inform you that your paper has been accepted for an oral presentation at MathAI 2026.

Your paper was evaluated through a rigorous two-stage review process involving both automated screening and expert review by members of the Program Committee. The reviewers recognized the quality and contribution of your work.

Presentation details:

- Format: Oral presentation (15–20 minutes + 5 minutes Q&A)
- Mode: You may present either in person (offline) at the conference venue in Sirius, Russia, or remotely via Zoom. Please indicate your preferred mode when confirming your participation.
- Conference dates: Marh 30 - April 3, 2026
- Website: https://mathai.club

Next steps:

1. Please confirm your participation and presentation mode by replying to this email mathai.club@yandex.ru no later than March 15, 2026 18:00 Moscow time.
2. If you plan to attend in person, the organizing committee will provide accommodation details separately.
3. Please prepare your final camera-ready manuscript according to the formatting guidelines available at https://mathai.club and upload it to OpenReview by March 15, 2026 18:00 Moscow time.

Should you have any questions regarding the program, logistics, or your presentation slot, please do not hesitate to contact us.

We look forward to your contribution to MathAI 2026.

With kind regards,

MathAI 2026 Program Committee
International Conference on Mathematics of Artificial Intelligence
https://mathai.club
OpenReview: https://openreview.net/group?id=mathai.club/MathAI/2026/Conference
Telegram: https://t.me/MathAI_club
Email: mathai.club@yandex.ru